# Role of c-Myc activating the transcription of PD-L1 in immune escape during benzene-induced malignant transformation of human lymphoblasts

Qihao Huang[1☉], Biyan Ye[1☉], Zhongming Ye[1], Shuyun Huang[1], Jiancheng Xue[1], Fengzhen Cui[1], Tikeng Jiang[1,2], Lei Sun[1,3], Yutao Zeng[1], Shaoying Wang[1], Yuting Chen[1]*, Huanwen Tang[1]*

**1** School of Public Health, the First Dongguan Affiliated Hospital, Guangdong Medical University, Dongguan, Guangdong, China, **2** Dongguan Maternal and Child Health Hospital, Dongguan, Guangdong, China, **3** The First Dongguan Affliated Hospital of Guangdong Medical University, Dongguan, Guangdong, China

☉ These authors contributed equally to this research.
* yutingchen@gdmu.edu.cn (YC); thw@gdmu.edu.cn (HT)

## Abstract

Benzene is a widely used industrial raw material. Benzene and its metabolite hydroquinone are believed to be related to the occurrence of benzene-related leukemia. Epidemiological studies have revealed the link between benzene exposure and blood system tumors, but the mechanism of benzene exposure and blood system tumor immune escape has not been fully confirmed. Experiments are divided into in vitro experiments and in vivo experiments. In the in vivo experimental part, we constructed an animal model of chronic benzene exposure and a C57BL/6 tumor-like animal model. Through qRT-PCR, Western blot and Tumor formation experiment in C57BL/6 mice, we verified that chronic benzene exposure caused malignant lesions in mice. In in vitro experiments, we built an in vitro malignant transformation model. Through molecular biology experiments 9/3/20259/3/20259/3/20259/3/2025such as CCK-8, soft agar cloning, and Transwell experiments were conducted to assess the malignant transformation potential of HQ19 and PBS19, cell vitality, malignant transformation potential and malignant migration ability were detected respectively. In order to study whether the benzene metabolite HQ activates c-Myc transcription activity, we used experiments such as double fluoresin enzyme reporting genes to confirm that c-Myc directly binds to PD-L1 promoter to drive its transcription. This study confirmed that in the process of HQ induced malignant transformation of human lymphocytes TK6, c-Myc regulates the molecular mechanism of PD-L1-mediated immune escape through transcription.

**Data availability statement:** All relevant data are within the manuscript and its Supporting Information files.

**Funding:** This research was supported by grants from the Guangdong Basic and Applied Basic Research Foundation (2023A1515140169), Guangdong Provincial University Key Platform Featured Innovation Project (2020KTSCX048), National Natural Science Foundation of China (82073582), Research Project of Guangdong Provincial Administration of Traditional Chinese Medicine (20242044), Undergraduate Innovation and Entrepreneurship Education Base Project of Guangdong Medical University (JDXM2024048).

**Competing interests:** Has been submitted to other materials.

# 1 Introduction

Benzene, a fundamental raw material in petrochemicals, is a colorless liquid with a characteristic aromatic odor [1]. Classified by the International Agency for Research on Cancer (IARC) as a Group 1 carcinogen, long-term benzene exposure impairs the hematopoietic system, potentially leading to leukemia and other hematological malignancies in severe cases [2]. Hydroquinone (HQ), a major benzene metabolite, is a white crystalline powder exhibiting potent immunosuppressive activity and immune escape potential. It mediates tumor cell evasion of immune surveillance by inhibiting inflammatory responses, modulating immune checkpoint molecule expression, and remodeling the immune microenvironment [3–5].

Immune escape denotes the phenomenon whereby tumor cells evade or suppress immune recognition and attack through orchestrated biological mechanisms, enabling persistent survival within the host [6]. During this process, tumor cells deploy diverse self-regulated strategies to circumvent immune attack, with transcription factors serving as key regulators. These factors modulate tumor cell immunogenicity and promote the formation of an immunosuppressive microenvironment by regulating gene expression, thereby facilitating tumor cell proliferation in vivo.

Transcription factors are proteins that specifically bind to gene promoter regions or regulatory elements, governing gene transcription by modulating RNA polymerase activity [7]. c-Myc, a prototypical transcription factor of the Myc gene family [8], exerts pleiotropic roles in cellular processes, including the regulation of proliferation, differentiation, apoptosis, and metabolism [9]. It is well established that c-Myc is critically involved in carcinogenesis, tumor progression, invasion, metastasis, and drug resistance. This master regulator governs not only cell fate decisions but also shapes the tumor immune microenvironment. As such, c-Myc represents a promising therapeutic target in oncology. Emerging evidence highlights a mechanistic link between c-Myc and PD-L1 expression [10], whereby c-Myc promotes immune escape by transcriptionally upregulating PD-L1 [11,12]. PD-L1 inhibits T-cell cytotoxicity by binding to PD-1 on T cells, enabling cancer cells to evade immune surveillance [13].

The regulatory mechanisms of c-Myc and PD-L1 have gradually attracted attention, but their roles and mechanisms in benzene-induced leukemia remain unclear and require in-depth investigation. Therefore, our study aims to validate immune escape-related gene signatures and elucidate the transcriptional regulatory network between c-Myc and PD-L1, illuminating the role of immune escape pathways in benzene-induced leukemia. This research seeks to provide novel molecular targets and a translational scientific basis for preventing benzene-associated hematological malignancies.

# 2 Materials and methods

## 2.1 Cell culture and benzene metabolite treatment

Human normal lymphoblastoid cell lines (TK6 cells, purchased from Shanghai Chiqiang Biology, Shanghai, China) were cultured in DMEM medium containing 10% fetal bovine serum and maintained in a constant temperature incubator at 37°C with

5% $CO_2$.The experimental group was treated with 20 µM hydroquinone (HQ) (purity > 99%, Sigma- Aldrich, USA) for 19 consecutive weeks (medium was changed every 3 days to maintain the exposure concentration), and the control group was treated with an equal volume of PBS. Cells were respectively named HQ19 and PBS19 cells. The dose of HQ was selected based on the concentration that down-regulated cell viability by 25% in the CCK-8 experiments.

## 2.2 RNA extraction and real-time quantitative PCR

Total RNA was extracted from cells and mouse peripheral blood using RNAiso reagent (TAKARA, Japan). RNA concentration and purity were assessed by measuring the $A_{260}$/$A_{280}$ ratio (1.8–2.0) with a NanoDrop 2000 (Thermo, USA). cDNA was synthesized by reverse transcriptionSeville, Wuhan, China) using 1 µg of total RNA, and mRNA expression levels of target genes were detected by qRT-PCR. Reaction conditions: 95°C for 30 s; 40 cycles (95°C for 5 s, 60°C for 34 s). Data were analyzed by the $2^{-\Delta\Delta CT}$ method and normalized to GAPDH as the internal reference gene.

## 2.3 Protein immunoblot analysis

Cells and mouse peripheral blood/tissues were lysed using RIPA buffer(CST, Danvers, Massachusetts, USA) containing protease inhibitors to extract protein samples. Protein concentrations were determined using a BCA protein assay kit. Proteins (30 µg) were separated by electrophoresis, transferred to PVDF membranes, blocked with 5% skimmed milk, washed with TBST, and then incubated with primary antibodies: anti-GAPDH (dilution ratio 1:8,000), anti-PD-L1 (dilution ratio 1:5,000), anti-c-Myc (dilution ratio 1:5,000), TP53 (dilution ratio 1:5,000), PD-1 (dilution ratio 1:1,000), and CTLA4 (dilution ratio 1:1,000). GAPDH was used as the internal reference protein. Gray scale analysis was performed using ImageJ.

## 2.4 Bioinformatics analysis

Binding sites of c-Myc protein to the upstream promoter sequence of the PD-L1 gene were predicted using the JASPAR database(https://jaspar.elixir.no/).

## 2.5 Immunofluorescence detection of c-Myc localization

Cells were fixed with 4% paraformaldehyde, permeabilized with 0.1% Triton X-100, and blocked with 5% BSA for 1h. Cells were sequentially incubated with c-Myc primary antibody (overnight at 4°C) and a fluorescently labeled secondary antibody (1h at room temperature in the dark), followed by nuclear staining with DAPI.

## 2.6 Dual luciferase reporter gene assay

The PD-L1 promoter region (−2000 to +100 bp) was cloned into the pGL4-Basic plasmid vector to design wild-type (WT, containing sites 84–95) and mutant (mutated sites 84–95) reporter plasmids. The c-Myc overexpression plasmid and pRL-TK internal reference plasmid were co-transfected using Lipofectamine 3000. Luciferase activity was detected 48h after transfection according to the Dual-Luciferase kit instructions.

## 2.7 In vivo tumor formation experiment

Thirty SPF-grade C57BL/6 mice (6–8 weeks old) were purchased from Zhuhai BesTest Bio-Tech Co., Ltd. All animal experiments were approved by the Experimental Animal Ethics Committee of Guangdong Medical University (Approval No.: GDY2104067) and were conducted in strict accordance with the NIH Guide for the Care and Use of Laboratory Animals and the 3R (Replacement, Reduction, Refinement) principles. The duration of the experiment was 21 days after tumor transplantation. A total of 30 mice were used in the experiment, and all were euthanized at the end of the study; no animals were found dead before the scheduled euthanasia. For euthanasia, mice were first anesthetized by

intraperitoneal injection of tribromoethanol (350 mg/kg, reach the anesthetic dose). After anesthesia, euthanasia was performed by cervical dislocation, and each mouse was sacrificed within approximately one minute.

Animal experiments were randomly divided into 5 groups (n = 6): PBS19, HQ19, HQ19 + c-Myc inhibitor (concentration 80 μM), HQ19 + PD-L1 inhibitor (concentration 20 μM),and HQ19 + c-Myc inhibitor + PD-L1 inhibitor combined treatment groups. A cell suspension (100 μL containing $5 \times 10^5$ cells) was subcutaneously injected into the right dorsal side of each mouse.

## 2.8  Statistical Analysis

All experiments were repeated three times and the results are expressed as means ± SE of experiments. Significant differences between groups were determined by one-way ANOVA and *t*-test. and a p-value of <0.05 was considered statistically significant.

## 3.  Result

### 3.1  Malignant transformation of TK6 cells under long-term HQ treatment

Cell proliferation, migration, and colony formation abilities were evaluated using CCK-8 (Fig 1A), transwell assay (Fig 1B), and soft agar colony formation assay (Fig 1C). The HQ19 group exhibited significantly higher activity in all assays compared to the PBS19 group, confirming successful establishment of the malignant transformation model. Transcriptome sequencing differential gene analysis (heatmap in Fig 1D) showed altered expression of immune escape-related genes in HQ19 cells versus PBS19. GEPIA2 database analysis (Fig 1E) revealed elevated TP53 expression in acute myeloid leukemia (AML). TCGA data further indicated differential expression of MYC, PD-L1, and TP53 in leukemia patients. Survival curve analysis (Fig 1F~1H) showed that leukemia patients with high MYC, PD-L1, or TP53 expression had lower 5-year survival rates.

### 3.2  Alterations in immune escape and malignant genes in TK6 cells after long-term HQ treatment

qRT-PCR and Western blot were used to analyze immune escape-related gene expression in HQ19 and PBS19 cells. PD-L1 (Fig 2A), c-Myc (Fig 2B), IL-10 (Fig 2D), and TP53 (Fig 2E) showed upregulated RNA levels in HQ19, while CD86 (Fig 2C) was downregulated. Additional negative immune escape regulators were screened (Fig 2F). Western blot analysis of protein levels (Fig 2G) showed that TP53 protein decreased in HQ19 (Fig 2H), possibly due to gene deacetylation (discrepant with RNA levels), whereas c-Myc (Fig 2I) and PD-L1 (Fig 2J) proteins were upregulated.

### 3.3  c-Myc acts as an upstream regulator of PD-L1 and modulates PD-L1 expression

To explore the c-Myc/PD-L1 regulatory axis, HQ19 cells were treated with c-Myc or PD-L1 inhibitors, followed by qRT-PCR and Western blot analysis. c-Myc inhibitor treatment dose-dependently reduced c-Myc and PD-L1 RNA levels (Fig 3A~B), while PD-L1 inhibitor decreased PD-L1 RNA (Fig 3C) without affecting c-Myc (Fig 3D). Western blot assay detected c-Myc and PD-L1 protein changes upon c-Myc inhibitor treatment (Fig 3E), with quantitative analysis showing that 80 μM c-Myc inhibitor reduced c-Myc (Fig 3F) and PD-L1 (Fig 3G) protein levels. Western blot analysis was performed to detect c-Myc and PD-L1 protein level changes following PD-L1 inhibitor treatment (Fig 3H). Quantitative analysis revealed that 20μM PD-L1 inhibitor induced no significant decrease in c-Myc protein (Fig 3I), whereas PD-L1 protein showed a marked reduction (Fig 3J). These findings suggest a direct or indirect binding interaction between c-Myc and PD-L1, with c-Myc acting as an upstream regulator of PD-L1.

### 3.4  Transcription factors c-Myc and PD-L1 have a binding site on 84–95, with a direct regulatory relationship

To identify regulatory binding sites between transcription factor c-Myc and PD-L1, immunofluorescence (IF) was used to visualize c-Myc localization. IF results showed nuclear localization of c-Myc in both HQ19 and PBS19 cells (Fig 4A),

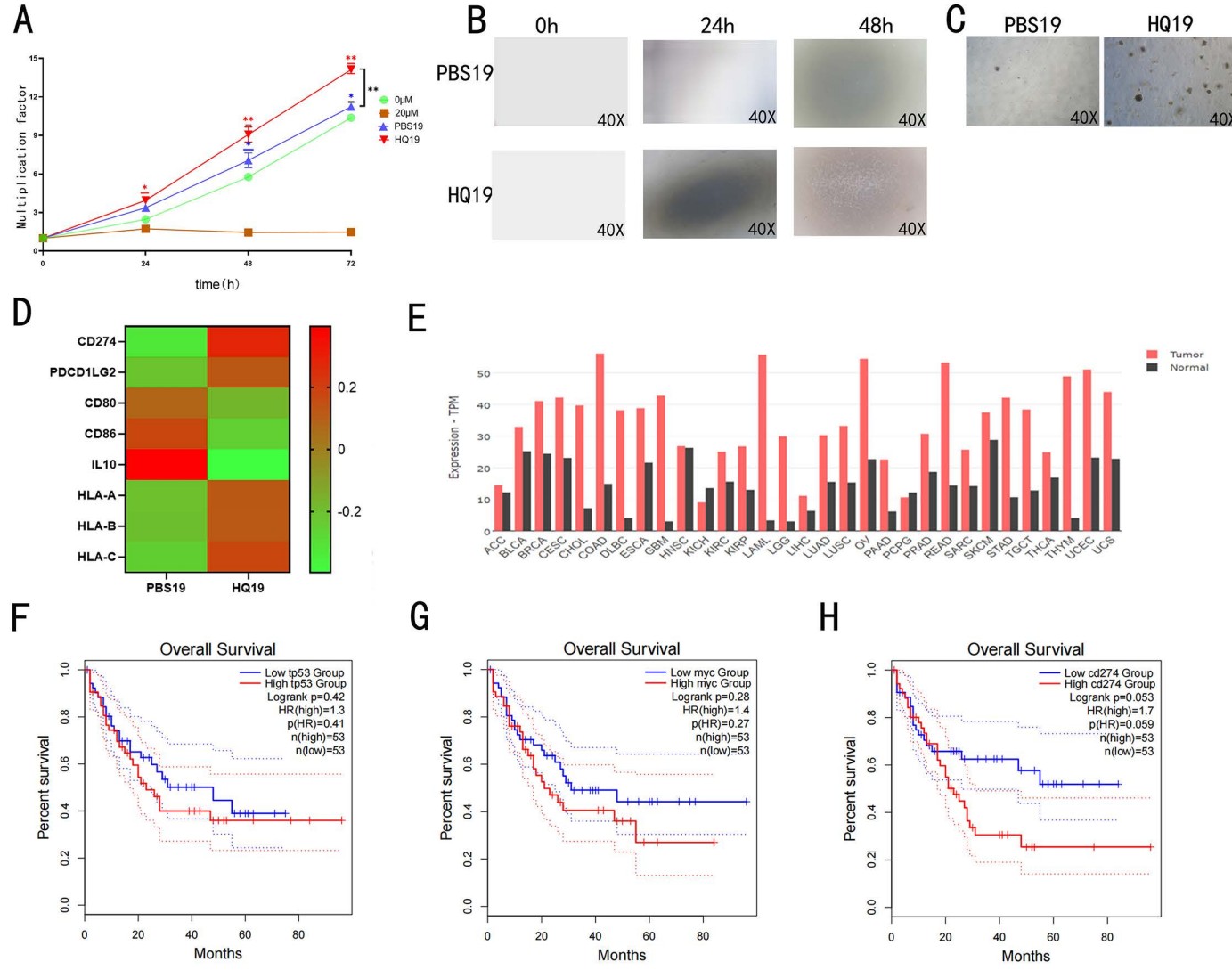

**Fig 1. Long-term HQ treatment induces malignant transformation of TK6 cells.** (A) Proliferation viability of HQ19 and PBS19 cells. (B) Invasion capacity of HQ19 and PBS19 cells. (C) Colony formation ability of HQ19 and PBS19 cells. (D) Differential gene expression heatmap analysis. (E) TP53 gene expression across various cancers. (F, G, H) Survival curves of cancer-related genes in leukemia patients. ns denotes no statistical significance, *: $P < 0.05$, **: $P < 0.01$.

with significantly enhanced fluorescence intensity in HQ19 compared to PBS19 (Fig 4B). Predicted binding sites of c-Myc on PD-L1 were mapped, and plasmids harboring gene silencing mutations at the 84–95 bp region were transfected into cells (Fig 4C). Dual luciferase reporter assays confirmed that c-Myc promoted PD-L1 expression (Fig 4D). Time-course qRT-PCR revealed increasing c-Myc RNA levels (Fig 4E) and gradual upregulation of PD-L1 RNA (Fig 4F), paralleling enhanced immune escape capacity. Western blot analysis showed time-dependent increases in c-Myc (Fig 4H) and PD-L1 (Fig 4I) protein levels, consistent with RNA expression trends (Fig 4G).

## 3.5 The immune response of T cells in the immune microenvironment

To elucidate the immunomodulatory mechanisms of T cells in the tumor microenvironment, we analyzed the RNA expression levels of critical genes, including c-Myc and PD-L1, in various leukemia cell lines. Using qRT-PCR, we quantified the

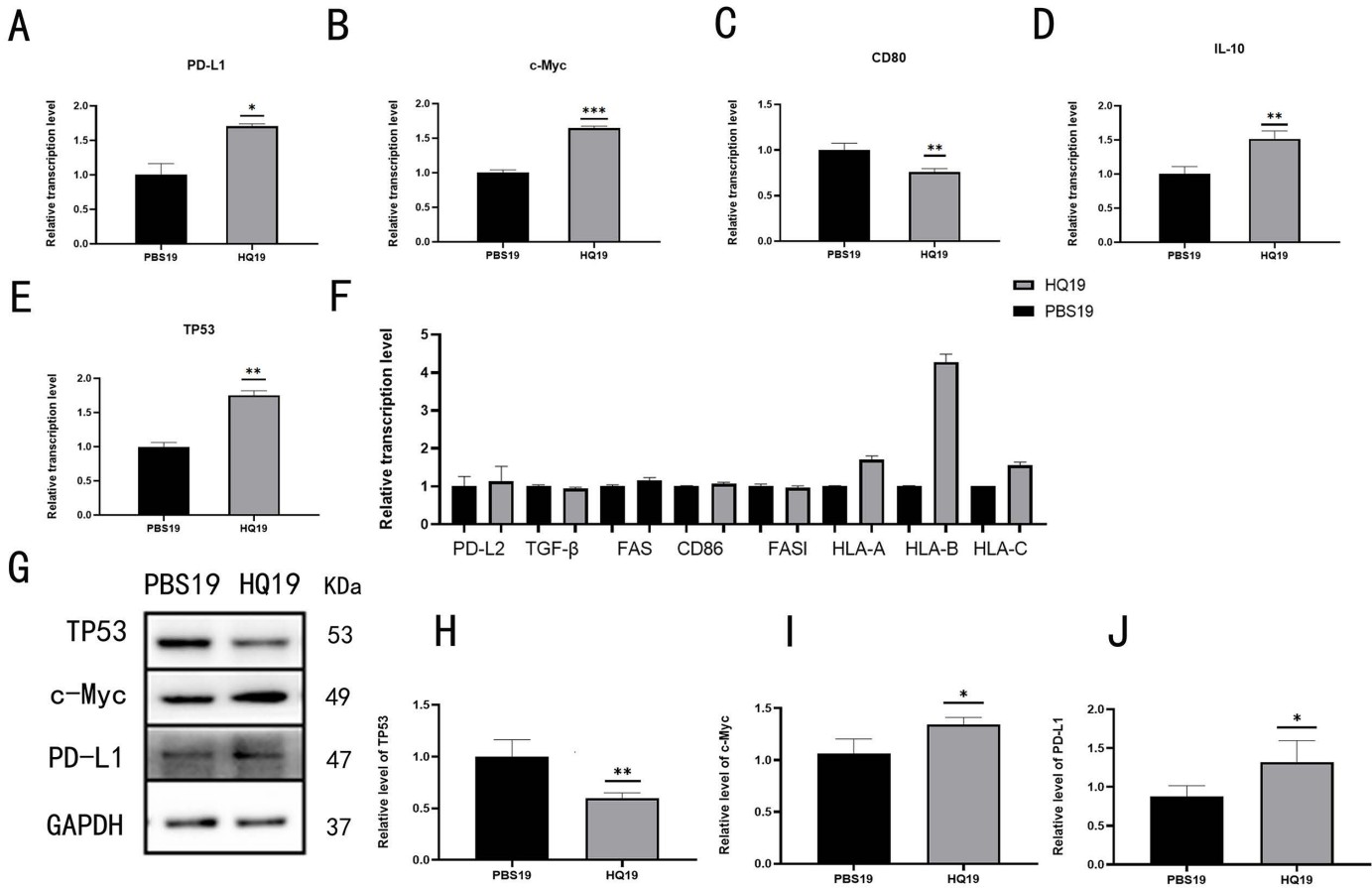

**Fig 2. Long-term HQ treatment alters immune escape and malignant genes in TK6 cells.** Transcriptional level of PD-L1. (B) Transcriptional level of c-Myc. (C) Transcriptional level of CD86. (D) Transcriptional level of IL-10. (E) Transcriptional level of TP53. (F) Other negative immune escape genes. (G) Protein expression levels of c-Myc, TP53, and PD-L1. (H) Protein expression level of TP53. (I) Protein expression level of c-Myc. (J) Protein expression level of PD-L1. ns denotes no statistical significance, *: $P<0.05$, **: $P<0.01$.

transcriptional levels of immune regulators, revealing that both c-Myc and PD-L1 were significantly upregulated in KG-1 cells compared to TK6 cells (Fig 5A~B). Western blot analysis further confirmed elevated protein expression of c-Myc and PD-L1 in KG-1 cells (Fig 5C~E). To model immune cell interactions within the microenvironment, we established a co-culture system of Jurkat T cells with KG-1 cells (using HQ19 as a positive control). Gene expression profiling demonstrated that co-culture with KG-1 significantly increased the RNA levels of immune checkpoint molecules CTLA4 and PD-1 in Jurkat cells compared to monoculture controls (Fig 5F~G). Western blotting validated these findings at the protein level, showing pronounced upregulation of CTLA4 and PD-1 in co-cultured Jurkat cells (Fig 5H~J). Collectively, these data suggest that KG-1 cells may promote the expression of immune checkpoint molecules in T cells by upregulating c-Myc and PD-L1.

### 3.6 Chronic benzene exposure causes malignant lesions in mice and HQ19 causes tumorigenesis in C57BL/6 mice

To investigate the carcinogenic effects of chronic benzene exposure, we established a mouse model by exposing C57BL/6 mice to benzene (0 or 20 mg/m³) via dynamic inhalation for 12 months. Following sacrifice, whole blood was collected for qRT-PCR and Western blot analysis of immune escape-related genes. Compared to controls, benzene-exposed

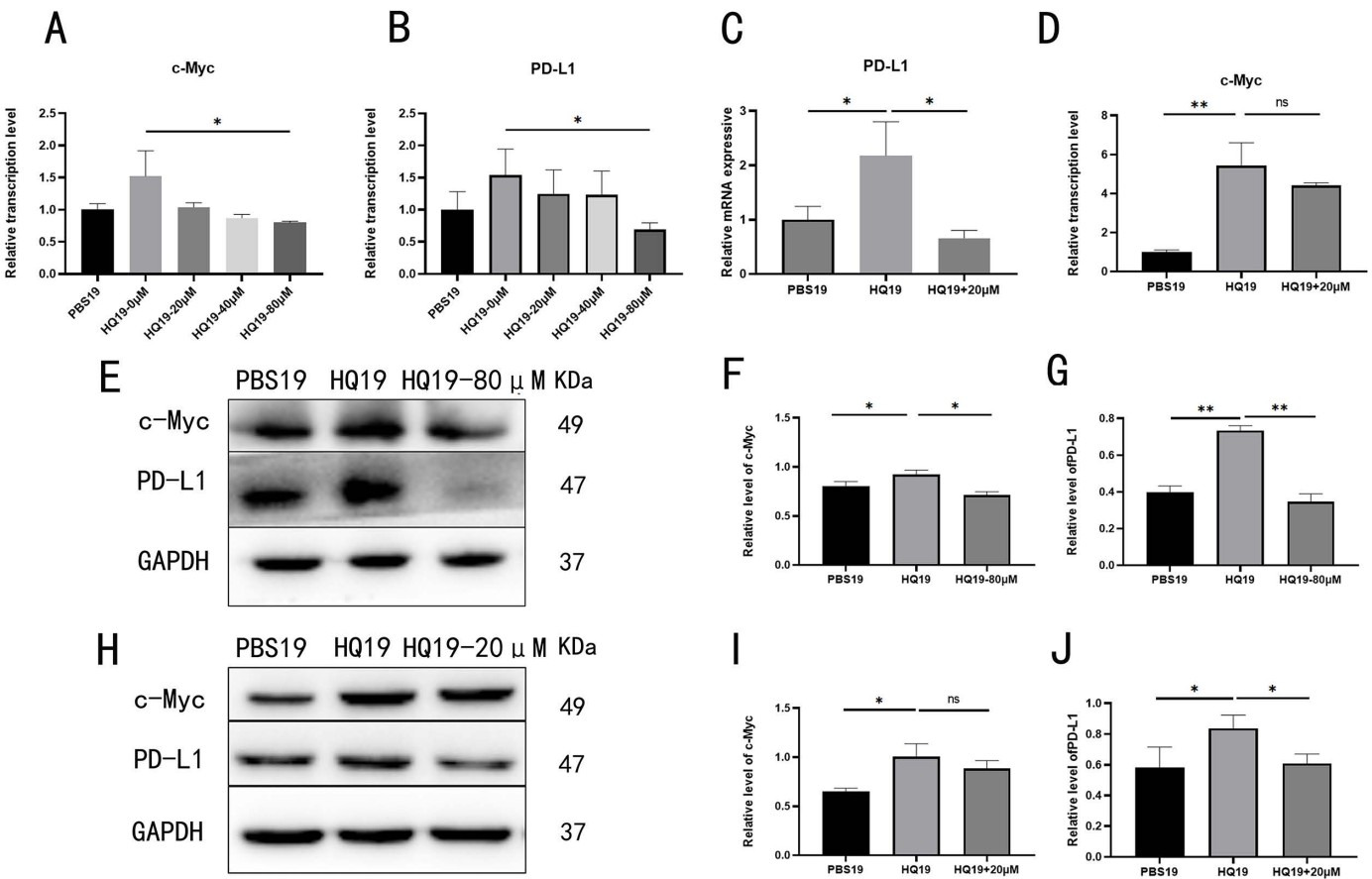

**Fig 3. PD-L1 expression is regulated by its upstream target gene, c-Myc.** (A) Transcriptional level of c-Myc. (B) Transcriptional level of PD-L1. (C) Transcriptional level of PD-L1. (D) Transcriptional level of c-Myc. (E) Protein expression levels of c-Myc and PD-L1. (F) Protein expression level of c-Myc. (G) Protein expression level of PD-L1. (H) Protein expression levels of c-Myc and PD-L1. (I) Protein expression level of c-Myc. (J) Protein expression level of PD-L1. ns denotes no statistical significance, *: $P < 0.05$, **: $P < 0.01$.

mice exhibited significantly higher RNA levels of PD-1, c-Myc, PD-L1, and CTLA4, whereas TP53 RNA showed no significant change (Fig 6A~E). Western blot analysis of blood samples further confirmed elevated protein expression of PD-1 (Fig 6G), c-Myc (Fig 6I), PD-L1 (Fig 6J), and CTLA4 (Fig 6K) in exposed mice, while TP53 protein was significantly downregulated (Fig 6H). These results indicate that chronic benzene exposure dysregulates immune escape pathways in vivo.

To assess the oncogenic potential of HQ19, we performed tumorigenesis assays in C57BL/6 mice using five treatment groups: PBS19, HQ19, HQ19+c-Myc inhibitor, HQ19+PD-L1 inhibitor, and HQ19+combined inhibitors (Fig 6L). Compared with the PBS19 control group, the tumors in the HQ19 group were significantly larger (Fig 6M). Treatment with either inhibitor alone or in combination significantly reduced tumor burden, demonstrating that HQ19 induced malignancy is dependent on c-Myc/PD-L1 signaling. Collectively, these findings establish that chronic benzene exposure promotes immune escape and malignancy in vivo, and that HQ19 driven tumorigenesis can be mitigated by targeting c-Myc/PD-L1 pathways.

## 4 Discussion

Benzene, a well-established carcinogen, is associated with leukemia development and various hematological malignancies following chronic exposure [14–16]. Its hematotoxic effects are primarily mediated through reactive metabolites (e.g.,

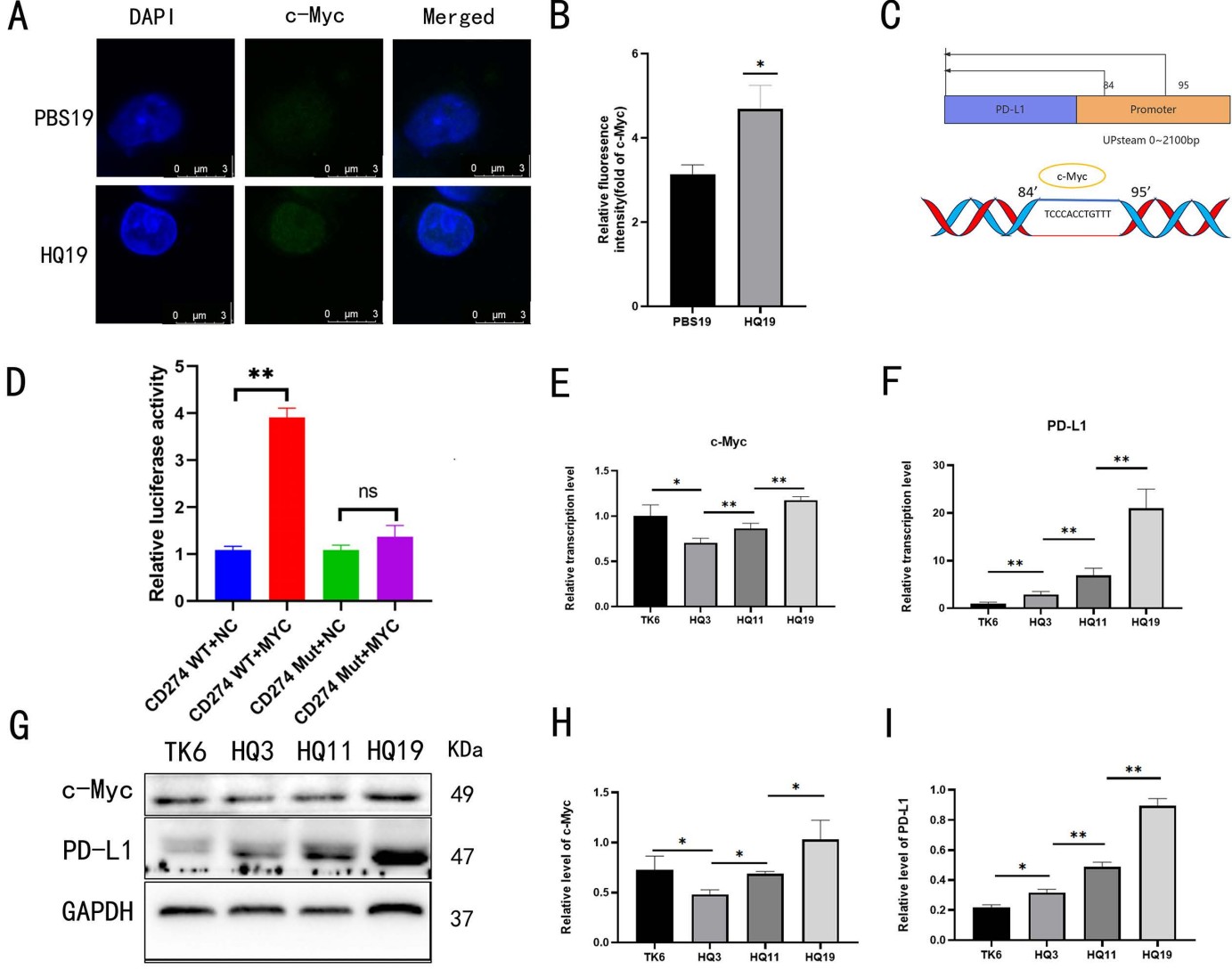

**Fig 4. Transcription factor c-Myc binds to PD-L1 at 84-95 site, establishing a direct regulatory relationship.** (A) c-Myc localization in HQ19 and PBS19. (B) c-Myc fluorescence intensity in HQ19 and PBS19. (C) Predicted binding sites of c-Myc and PD-L1. (D) Relative fluorescence intensity comparison between WT and MUT groups. (E) Time-course transcriptional level of c-Myc. (F) Time-course transcriptional level of PD-L1. (G) Time-course protein expression levels of c-Myc and PD-L1. (H) Time-course protein expression level of c-Myc. (I) Time-course protein expression level of PD-L1. ns denotes no statistical significance, *: $P<0.05$, **: $P<0.01$.

phenol and hydroquinone) that form covalent adducts with DNA and cellular proteins [17], inducing genotoxic damage and functional impairments in hematopoietic cells [18,19]. Hydroquinone, as one of the important intermediates in the process of benzene metabolism, has a toxic role in the hematotoxicity. Studies have shown that HQ inhibits cell proliferation, induces apoptosis, and may provide a pathogenic basis for the development of leukemia through the effects of hematopoietic stem precursor cell self-renewal [20,21]. However, although TK6 cells possess wild-type p53 function and relatively high genomic stability, the presence of EBV may cause certain interference to the genomic stability of the cells [22]. Benzene is a known genotoxic substance that can cause DNA damage and gene mutations. In TK6 cells, EBV infection may interact with the genetic toxicity of benzene, affecting the sensitivity of the cells to the gene mutations induced by benzene

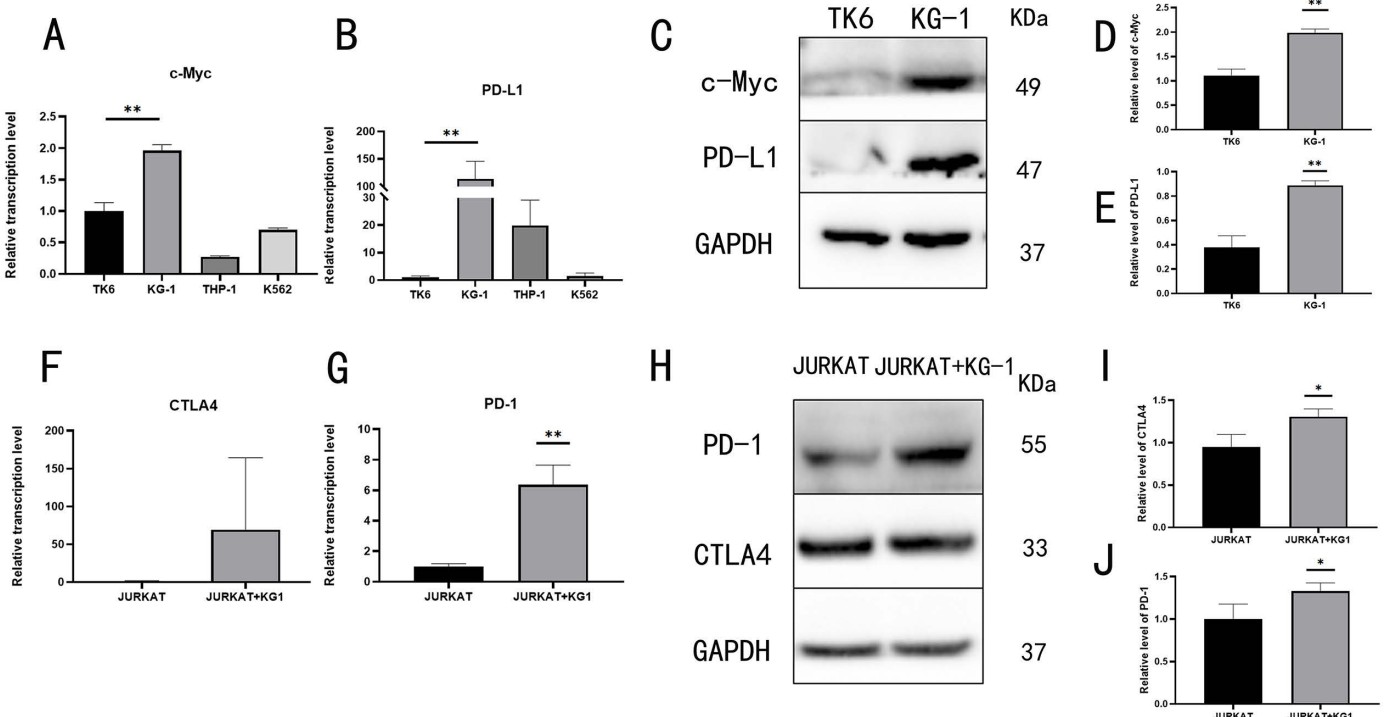

**Fig 5. Immune response of T cells in the immune microenvironment.** (A) Transcriptional level of c-Myc in different leukemia cell lines. (B) Transcriptional level of PD-L1 in different leukemia cell lines. (C) Protein expression levels comparing KG-1 with TK6. (D) c-Myc protein expression level comparing KG-1 with TK6. (E) PD-L1 protein expression level comparing KG-1 with TK6. (F) Transcriptional level of CTLA4. (G) Transcriptional level of PD-1. (H) T cell protein expression levels after co-culture of Jurkat and KG-1. (I) Protein expression level of CTLA4. (J) Protein expression level of PD-1. ns denotes no statistical significance, *: $P < 0.05$, **: $P < 0.01$.

[23]. There is a certain connection between EBV infection and cell transformation. In TK6 cells, the presence of EBV may enhance the cell transformation process induced by benzene. Some proteins encoded by EBV can activate intracellular signaling pathways, promoting cell proliferation and survival, which may work synergistically with the carcinogenic effect of benzene to accelerate the malignant transformation of cells [24].

While previous investigations have primarily focused on benzene-induced lymphoblastoid cell transformation and associated immune evasion mechanisms, our study specifically elucidates the novel regulatory axis between c-Myc and PD-L1 in immune escape processes, providing critical insights for developing targeted therapies against benzene-related leukemia.

Immune escape is a phenomenon whereby tumor cells evade immune-mediated killing and survive within the host via diverse mechanisms [25]. Tumor cells mediate immune escape by expressing immune checkpoint molecules (e.g., PD-L1) or modulating the tumor microenvironment [26,27].

HQ induced cell proliferation and malignant transformation assays demonstrated that prolonged HQ treatment significantly promoted the proliferative and invasive capacities of TK6 cells (Fig 1A). Soft agar colony formation and Transwell migration assays further validated that HQ treated cells exhibited enhanced proliferative and invasive phenotypes (Fig 1BC). Collectively, these findings indicate that HQ drives cellular malignant transformation and alters biological behavior. Transcriptome sequencing revealed dysregulated expression of immune escape-related genes in HQ induced TK6 cells (Fig 1D), suggesting that HQ transformed cells evade immune surveillance via immune checkpoint imbalance. Furthermore, GEPIA2 and TCGA database analyses showed elevated c-Myc, PD-L1, and TP53 expression across multiple

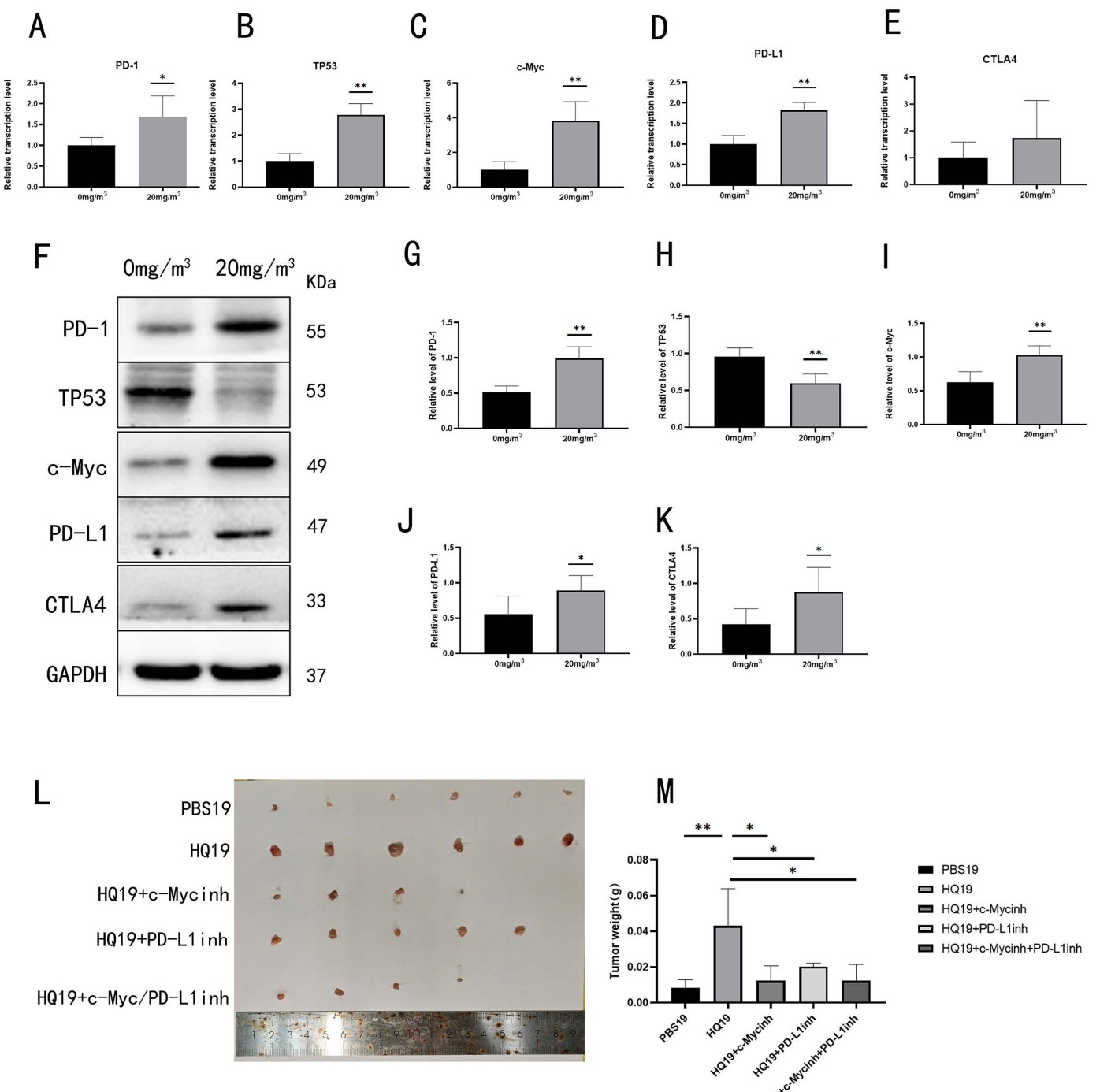

**Fig 6. Chronic benzene exposure induces malignant lesions in mice and HQ19 promotes tumorigenesis in C57BL/6 mice.** (A) Transcriptional level of PD-1. (B) Transcriptional level of TP53. (C) Transcriptional level of c-Myc. (D) Transcriptional level of PD-L1. (E) Transcriptional level of CTLA4. (F) Protein expression levels of immune escape-related genes in whole blood of mice after 12-month dynamic inhalation exposure. (G) PD-1 protein expression level. (H) c-Myc protein expression level. (I) TP53 protein expression level. (J) PD-L1 protein expression level. (K) CTLA4 protein expression level. (L) Tumor size of five gradient groups in tumorigenesis assay. (M) Tumor weight of five gradient groups in tumorigenesis assay. ns denotes no statistical significance, *: $P < 0.05$, **: $P < 0.01$.

cancer types, with leukemia patients exhibiting high gene expression displaying reduced 5-year survival rates (Fig 1E~H). These results further corroborate the critical roles of these genes in tumorigenesis and immune escape mechanisms.

The c-Myc/PD-L1 regulatory axis was investigated using multi-level experimental approaches. First, qRT-PCR and Western blot analyses showed that chronic HQ treatment significantly upregulated c-Myc and PD-L1 expression at both mRNA and protein levels (Fig 2). Pharmacological inhibition experiments revealed that c-Myc suppression dose-dependently reduced PD-L1 expression, whereas PD-L1 inhibition did not affect c-Myc levels—findings that establish c-Myc as an upstream regulator of PD-L1 (Fig 3).Dual luciferase reporter assays and gene silencing experiments further confirmed direct binding of c-Myc to the PD-L1 promoter at the 84–95 bp region. Specifically, mutation of this site abrogated c-Myc-mediated PD-L1 transactivation, demonstrating that c-Myc regulates PD-L1 expression through specific promoter binding (Fig 4C~D). Collectively, these results elucidate the critical role of c-Myc in HQ induced immune escape and offer mechanistic insights for downstream functional studies.

The study of the mechanism of tumor cells evading the body's immune surveillance is one of the key mechanisms explored in the immune escape mechanism. In this study, we found that HQ induced TK6 cells (HQ19) were able to significantly affect the immune response of T cells through co-culture experiments and related gene detection. Specifically, Jurkat T cells co-cultured with KG-1 cells showed upregulated PD-1 and CTLA-4 expression (Fig 5), indicating that HQ transformed cells suppress T cell function by enhancing immunosuppressive molecule expression (e.g., PD-L1).

To validate HQ induced malignant transformation and immune escape mechanisms, we established chronic benzene-exposed mouse models and C57BL/6 tumorigenesis assays (Fig 6L). Results showed that HQ transformed cells exhibited enhanced malignant potential and promoted tumor progression via immune escape pathways.

Despite these findings, the study has several limitations. First, reliance on in vitro models and short-term animal experiments, combined with a limited sample size in C57BL/6 tumorigenesis assays, may compromise result generalizability. Long-term carcinogenic mechanisms under chronic benzene exposure require further investigation. Second, while the c-Myc/PD-L1 regulatory axis was validated, specific molecular mechanisms (e.g., epigenetic regulation, signaling cross-talk) warrant deeper exploration.

## 5 Conclusion

The benzene metabolite hydroquinone (HQ) facilitates immune evasion and malignant transformation of TK6 cells.The transcription factor c-Myc can regulate the immune escape effect induced by activating the transcription of PD-L1. From this perspective, if the role of c-Myc in immune evasion can be confirmed through future clinical experiments, then targeting the c-Myc/PD-L1 axis may serve as a potential therapeutic strategy for hematological disorders caused by benzene exposure.

## Supporting information

**S1 File. Original experimental data.part01. To access the complete raw data, readers should download all eight Zip-format compressed packages (about 600 MB) in full, followed by unzipping them.**
(RAR)

**S2 File. Original experimental data.part02.**
(RAR)

**S3 File. Original experimental data.part03.**
(RAR)

**S4 File. Original experimental data.part04.**
(RAR)

**S5 File. Original experimental data.part05.**
(RAR)

**S6 File. Original experimental data.part06.**
(RAR)

**S7 File. Original experimental data.part07.**
(RAR)

**S8 File. Original experimental data.part08.**
(RAR)

## Acknowledgments

The laboratory of Guangdong Medical University provided experimental equipment support.

## Author contributions

**Conceptualization:** Qihao Huang, Zhongming Ye, Yuting Chen, Huanwen Tang.

**Data curation:** Qihao Huang, Biyan Ye, Yutao Zeng.

**Formal analysis:** Qihao Huang, Biyan Ye, Yutao Zeng.

**Funding acquisition:** Yuting Chen, Huanwen Tang.

**Investigation:** Qihao Huang, Tikeng Jiang, Lei Sun, Yutao Zeng.

**Methodology:** Qihao Huang, Tikeng Jiang, Lei Sun, Shaoying Wang.

**Project administration:** Zhongming Ye, Tikeng Jiang, Lei Sun, Shaoying Wang, Huanwen Tang.

**Resources:** Zhongming Ye, Shaoying Wang, Yuting Chen, Huanwen Tang.

**Software:** Biyan Ye, Shuyun Huang, Fengzhen Cui, Shaoying Wang.

**Supervision:** Shuyun Huang, Jiancheng Xue, Fengzhen Cui.

**Validation:** Zhongming Ye, Shuyun Huang, Jiancheng Xue, Fengzhen Cui.

**Visualization:** Jiancheng Xue, Fengzhen Cui.

**Writing – original draft:** Qihao Huang.

**Writing – review & editing:** Yuting Chen, Huanwen Tang.

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
