## [Decision Letter · Decision Letter 0]

13 Aug 2025

Dear Dr. Tang,

Your manuscript was reviewed by two experts in the field. I also quickly reviewed the data section. Overall, this is a valuable contribution to science and valuable mechanistic insights into benzene-induced carcinogenesis through the c-Myc/PD-L1 pathway, but requires substantial revision before acceptance. You must address four critical areas: (1) **English language quality** - the manuscript contains numerous grammatical errors and scientific terminology inconsistencies that significantly impair readability requires professional editing; (2) **Statistical analysis** - the statistical methodology needs clearer description, sample size justification, and more rigorous presentation of significance testing results; (3) **Figure legends** must be substantially expanded to facilitate reader comprehension, including detailed descriptions of experimental conditions, sample sizes, and statistical methods used; and (4) **Critical cell line characterization** - TK6 is described as "human normal lymphoblastoid cell lines" but fail to address their origin and relationship to Epstein-Barr virus (EBV). TK6 cells contain EBV DNA. While TK6 cells remain appropriate for benzene toxicology studies due to their wild-type p53 function, genomic stability, and extensive validation in genotoxicity research, you must provide proper discussion including their derivation history and potential implications of any EBV relationship on their benzene-induced transformation studies. The scientific merit of identifying the c-Myc/PD-L1 regulatory axis is strong, but these fundamental presentation and methodological issues must be resolved to meet publication standards.

We look forward to receiving your revised manuscript.

Kind regards,

Luwen Zhang

Academic Editor

PLOS ONE

Journal Requirements:

2. Thank you for stating the following financial disclosure: [This research was supported by grants from National Natural Science Foundation of China (82073582 )、the Guangdong Basic and Applied Basic Research Foundation (2023A1515140169)、Guangdong Provincial University Key Platform Featured Innovation Project (2020KTSCX048)、Research Project of Guangdong Provincial Administration of Traditional Chinese Medicine (20242044)、Undergraduate Innovation and Entrepreneurship Education Base Project of Guangdong Medical University (JDXM2024048).]. 

3. Thank you for stating the following in your Competing Interests section: [Has been submitted to other materials].

4. In this instance it seems there may be acceptable restrictions in place that prevent the public sharing of your minimal data. However, in line with our goal of ensuring long-term data availability to all interested researchers, PLOS’ Data Policy states that authors cannot be the sole named individuals responsible for ensuring data access (http://journals.plos.org/plosone/s/data-availability#loc-acceptable-data-sharing-methods).

Reviewers' comments:

Reviewer's Responses to Questions

**Comments to the Author**

1. Is the manuscript technically sound, and do the data support the conclusions?

Reviewer #1: Yes

Reviewer #2: Yes

2. Has the statistical analysis been performed appropriately and rigorously?

Reviewer #1: No

Reviewer #2: Yes

3. Have the authors made all data underlying the findings in their manuscript fully available?

Reviewer #1: No

Reviewer #2: Yes

4. Is the manuscript presented in an intelligible fashion and written in standard English?

Reviewer #1: Yes

Reviewer #2: Yes

Reviewer #1: Results from this study did support the suggested conclusion that in the process of HQ-induced malignant transformation of human lymphocytes TK6, c-Myc regulates the molecular mechanism of PD-L1-mediated immune escape through transcription. However, the statistical analysis isn't elaborate enough as a thorough search was needed on the readers' part for specific statistics. Additionally, conclusions and discussion could be elaborated in more details.

Reviewer #2: The study offers promising insights into novel finding that c-Myc directly activates PD-L1 transcription, providing a new mechanism for immune evasion in benzene-induced malignant transformation. The study's robust methodology, large sample size, and meticulous analysis, which provided compelling evidence for the study. Moreover, the results obtained from the study are robust and provide strong support for the central hypothesis, effectively bridging a significant gap in understanding how environmental factors contribute to cancer and its ability to evade the immune system. However, upon reviewing the manuscript, the study is interesting, well-conceived and worthy of publication. The authors should address the following comments:

Lines 107-108. Provide company name, city, & country of Reverse transcriptase.

Line 114. Provide company name, city, & country of RIPA buffer.

Line 156. Error? Or change the bold “reach the anesthetic dose).

Figures: The resolution for all the figures needs to be fixed.

**Do you want your identity to be public for this peer review?** For information about this choice, including consent withdrawal, please see our Privacy Policy

Reviewer #1: No

Reviewer #2: **Yes: ** Conrad Chibunna Achilonu

---

## [Author Response · Author response to Decision Letter 1]

25 Aug 2025

Thank you. I have no other comments.

---

## [Editor Report · Decision Letter 1]

29 Aug 2025

Role of c-Myc activating the transcription of PD-L1 in Immune Escape during Benzene-Induced Malignant Transformation of Human Lymphoblasts

PONE-D-25-34481R1

Dear Dr. Tang,

We’re pleased to inform you that your manuscript has been judged scientifically suitable for publication and will be formally accepted for publication once it meets all outstanding technical requirements.

Kind regards,

Luwen Zhang

Academic Editor

PLOS ONE
---

## [Editor Report · Acceptance letter]

PONE-D-25-34481R1

PLOS ONE

Dear Dr. Tang,

I'm pleased to inform you that your manuscript has been deemed suitable for publication in PLOS ONE. Congratulations! Your manuscript is now being handed over to our production team.

Kind regards,

on behalf of

Dr Luwen Zhang

Academic Editor

PLOS ONE